# The critical role of ultra-low-energy vibrations in the relaxation dynamics of molecular qubits

E. Garlatti [1,2], A. Albino [3], S. Chicco[1], V. H. A. Nguyen[4], F. Santanni[3], L. Paolasini [5], C. Mazzoli [6], R. Caciuffo[7], F. Totti [3], P. Santini[1,2], R. Sessoli [3] ✉, A. Lunghi [4] ✉ & S. Carretta [1,2] ✉

Improving the performance of molecular qubits is a fundamental milestone towards unleashing the power of molecular magnetism in the second quantum revolution. Taming spin relaxation and decoherence due to vibrations is crucial to reach this milestone, but this is hindered by our lack of understanding on the nature of vibrations and their coupling to spins. Here we propose a synergistic approach to study a prototypical molecular qubit. It combines inelastic X-ray scattering to measure phonon dispersions along the main symmetry directions of the crystal and spin dynamics simulations based on DFT. We show that the canonical Debye picture of lattice dynamics breaks down and that intra-molecular vibrations with very-low energies of 1-2 meV are largely responsible for spin relaxation up to ambient temperature. We identify the origin of these modes, thus providing a rationale for improving spin coherence. The power and flexibility of our approach open new avenues for the investigation of magnetic molecules with the potential of removing roadblocks toward their use in quantum devices.

In the last years, molecular magnetism has been giving significant contributions to the second quantum revolution providing promising systems for strategic fields like quantum computing and simulation[1–5]. Indeed, magnetic molecules with two- or multi-level energy structure suitable to encode qubits or qudits were synthesized and proposed for new quantum architectures and schemes[6–21]. A pivotal step to improve the performance of molecular qubits is to reduce relaxation and decoherence due to phonons[22–26]. In particular, a key stage to achieve this result is to identify the vibrational modes responsible for this irreversible dynamics and devise recipes for the synthesis of improved systems.

A full comprehension of phonon-induced mechanisms in molecular nanomagnets (MNMs) requires a thorough and experimentally-assessed description of phonon modes and spin-phonon couplings coefficients. Indeed, only with such a solid starting point, it is possible to disentangle the complex correlations between phonon energies, their coupling to the spins, and their role in different decoherence processes, which is crucial to obtain a sound and consistent interpretation of experimental results. A synergistic approach combining cutting-edge experimental and theoretical techniques is therefore required to investigate phonons and their role in spin relaxation, namely: i) an experimental technique able to directly access phonon energies and polarization vectors in very-small-sized and [1]H-rich single-crystal samples, typical of molecular qubits; ii) state-of-the-art ab initio spin dynamics simulations, which must include an accurate description of phonon

[1]Dipartimento di Scienze Matematiche, Fisiche e Informatiche, Università di Parma and UdR Parma, INSTM, I-43124 Parma, Italy. [2]INFN, Sezione di Milano-Bicocca, gruppo collegato di Parma, I-43124 Parma, Italy. [3]Dipartimento di Chimica 'Ugo Schiff', Università Degli Studi di Firenze and UdR Firenze, INSTM, I-50019 Sesto Fiorentino, Italy. [4]School of Physics, AMBER and CRANN Institute, Trinity College, Dublin 2, Ireland. [5]ESRF - The European Synchrotron Radiation Facility, F-38043 Grenoble, Cedex 09, France. [6]National Synchrotron Light Source II, Brookhaven National Laboratory, Upton, NY 11973, USA. [7]INFN, Sezione di Genova, I-16146 Genova, Italy. ✉e-mail: roberta.sessoli@unifi.it; lunghia@tcd.ie; stefano.carretta@unipr.it

modes across the entire Brillouin zone and spin-phonon couplings coefficients.

Here we choose the well-characterized and radiation-robust [VO(TPP)] complex (VO = vanadyl, TPP = tetraphenylporphyrinate)[27] as a benchmark to show the capabilities of a multi-technique approach with all these characteristics. [VO(TPP)] is also a very promising molecular qubit: It allows simultaneous coherent manipulation of both electronic and nuclear spins[20,28] and it forms dimeric species where the two electronic spins are distinguishable and exchange-coupled to implement quantum gates[29]. To fulfill point (i), we exploit Inelastic X-ray Scattering (IXS). This technique has never been used before to address magnetic molecules but has several advantages with respect to more traditional spectroscopy techniques, such as inelastic neutron scattering. Crucially, IXS makes it possible to investigate very small single crystals, typical of MNMs, has energy-independent resolution and a very small background. In this work, we present a direct measurement of phonons in a molecular qubit obtained with IXS. The unique capabilities of the ID28 beamline at ESRF enable the measurement of acoustic and optical branches of [VO(TPP)] along different directions in the reciprocal space, probing both their energies and polarization vectors. Our results demonstrate that IXS has the sensitivity and the power to become the new technique of choice to investigate phonons and vibrations in molecular qubits and in MNMs in general. In particular, we find ultra-low-energy optical phonon modes at about 1-2 meV. Even if low-energy optical modes are a typical feature of molecular crystals of MNMs[30–33], [VO(TPP)], to the best of our knowledge, sets a new record, which also outdoes other ordered systems with low-energy optical phonons, like thermoelectric materials[34–36].

Moving to point (ii), IXS results are compared with state-of-the art periodic Density Functional Theory (pDFT) calculations of phonon energies and polarization vectors. Experimental and simulated IXS cross-sections are in very good agreement, validating the DFT results as the starting points for further analysis. DFT simulations also confirm the presence of ultra-low-energy optical modes in [VO(TPP)] phonon dispersions, which can deeply affect coherence time. The dominant role of these modes in both low- and high-temperature spin relaxation of MNMs was recently suggested by neutron scattering experiments and ab initio simulations[25,26,31–33,37–43]. Despite these studies provide a picture of spin relaxation that substantially differ from the canonical one based on the Debye model, a direct demonstration of the role played by non-Debye low-energy vibrations is yet to be achieved. In this work, spin dynamics simulations based on DFT are also exploited to provide a full picture of phonon-induced relaxation and decoherence in the benchmark molecular qubit [VO(TPP)]. We performed a neural-network based interpolation of the DFT calculations to estimate the spin-phonon couplings coefficients in [VO(TPP)], revealing that these low-energy optical phonons also possess very strong couplings to the spin. Finally, we demonstrate that ultra-low-energy vibrations are responsible for magnetic relaxation up to ambient temperature. We also report calculations of phonon-induced decoherence of MNMs including the important pure dephasing contribution of two-phonon processes. Finally, by comparing periodic and single-molecule DFT calculations, we also found that low-energy optical phonons in [VO(TPP)] are mainly associated with intra-molecular vibrations of specific chemical groups, pinpointing possible synthetic strategies towards the improvement of the spin coherence in this important class of molecules.

## Results

### Unveiling phonons with Inelastic X-ray Scattering

The very first results on the characterizations of phonons in molecular qubits and MNMs have been obtained only very recently with the Inelastic Neutron Scattering (INS) technique[31]. Despite the capabilities of the new generation of high-flux neutron spectrometers, this technique still requires very large high-quality single crystals to enable the investigation of phonon dispersions and polarization vectors along different directions in the reciprocal space. The very-small size of molecular crystals therefore typically prevents the use of INS to access phonon dispersions. Here we demonstrate that a cutting-edge technique allowing us to overcome this hurdle is high-resolution IXS[44,45], the possibility to use very small samples (of the order of 1 mm³) being its main advantage. Further advantages of IXS are due to the orders of magnitude difference between the incident hard X-rays ($E_i > 10$ keV) and the energy scale of interest when investigating phonons and vibrations (1−10 meV, $\Delta E/E_i$ ~ $10^{-7}$). This leads to an energy-independent resolution and to a complete decoupling between energy and momentum transfer, the latter being defined only by the scattering angle. Furthermore, IXS is essentially a background-free technique, since the incoherent cross-section for X-rays involve larger energy transfers (>eV) with respect to the energy window of interest, and multiple scattering is negligible[44]. Thus, deuterated samples are not required in IXS experiments, contrary to INS ones, where the large incoherent cross-section of hydrogen atoms, abundant in MNMs, can mask the coherent phonon signal.

Thanks to these specific features of the IXS technique, we were able to measure a [VO(TPP)] single-crystal with a size of the order of $1 \times 1 \times 0.5$ mm³ and all the data were obtained by measuring just one sample with no radiation damage (same Bragg peaks before and after the experiment and no induced colour centres). The experiment was performed on ID28 at the European Syncrotron facility ESRF[46], one of the very few IXS beamlines providing the required resolution and line-shape profile suitable for performing phonon studies in MNMs. Its unique trade-off between the high-energy resolution (up to $\delta E$ ~ 1.5 meV) and the incident flux enables the measurement of inelastic spectra in the energy range of interest for investigating low-energy phonons in molecular qubits. Moreover, a wide range of accessible momentum transfer allows the exploration of a wide section of the reciprocal space over specific symmetry directions in both longitudinal and transverse configurations. Measurements were performed at room temperature, with two different X-ray incident energies and resolutions (see Methods for more details on ID28). By using the low-resolution configuration ($\delta E = 3.0$ meV), we performed a first exploration of [VO(TPP)] phonon modes with constant-Q energy scans up to 25 meV, then we switched to the highest resolution configuration yielding $\delta E = 1.5$ meV, in order to investigate low-energy phonons in selected portions of the reciprocal space. We explored [VO(TPP)] reciprocal space along the Γ−N, Γ−K$_x$ and Γ−K$_z$ symmetry directions (($h0h$), ($h00$) and ($00l$), respectively in conventional cell notation, see Methods and Supplementary Note 1 for more details).

Intensities as a function of energy for some representative **Q** values are reported in Fig. 1 for the Γ−N direction off the (0 0 6) reciprocal lattice point (panels a, b). Longitudinal scans along the Γ−K$_x$ and Γ−K$_z$ directions in the same Brillouin Zone (BZ) are shown in panels (d,e) and (g,h), respectively. In Fig. 2a−c we show data along the Γ−N direction off the (6 0 0) reciprocal lattice point and the transverse scans along Γ−K$_z$ and Γ−K$_x$ directions. Phonon energies extracted from the data over the whole explored **Q** range along these directions are also reported in Fig. 1c, f, i and Fig. 2d−f, superimposed to DFT calculations (*vide infra*). Low-resolution data demonstrate the presence of longitudinal acoustic modes along both Γ−K$_z$ and Γ−K$_x$ directions, while non-dispersive transverse and longitudinal optical phonon branches are clearly visible at about 8-10 meV and 15 meV along Γ−N and Γ−K$_z$ directions, respectively. Measurements along Γ−K$_z$ but with transverse **Q** configurations also allowed the measurements of optical phonons with ultra-low energy. A non-dispersive mode was in fact detected at about 2 meV (Fig. 2a, d), clearly emerging from the elastic line (panel a) and visible over the whole explored **Q** range/half BZ (panel d). To better resolve phonon modes down to very-low energies along other directions, we switched to the high-resolution configuration with $\delta E = 1.5$ meV.

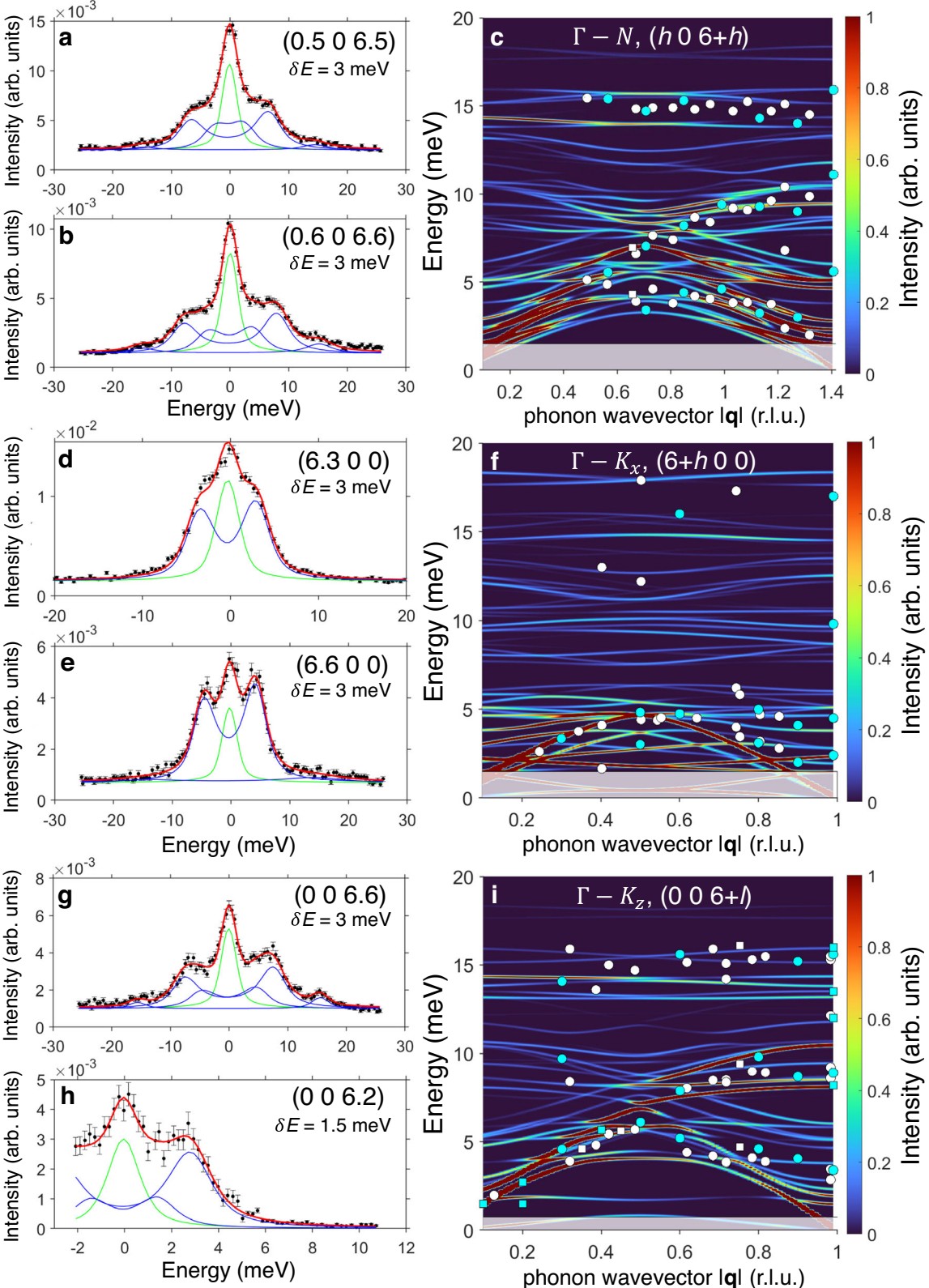

**Fig. 1 | Acoustic and optical phonons up to 20 meV.** ID28 data on [VO(TPP)] (black scatters, with error bars representing the SE) obtained at $T$ = 300 K and measured with resolutions $\delta E$ = 1.5, 3 meV as reported on each panel. **a**, **b** Show phonon excitations along the symmetry direction Γ–N at the (0.5 0 6.5) and (0.6 0 6.6) points in the reciprocal space, respectively; **d**, **e** display longitudinal phonon modes along the symmetry direction Γ–$K_x$ at the (6.3 0 0) and (6.6 0 0) points, respectively, while **g**, **h** report data of longitudinal scans along the symmetry direction Γ–$K_z$ at the (0 0 6.6) and (0 0 6.2) points. Solid red lines are results of a fit obtained with FIT28 (see Methods), comprising the excitation signals (blue lines)

and the elastic one (green line). In **c**, **f**, **i** we show the IXS cross-section (colour map) simulated with pDFT phonon energies and polarization vectors along Γ–N, long-itudinal Γ–$K_x$ and longitudinal Γ–$K_z$ directions. Cyan dots/squares are experimental IXS excitation energies extracted from the complete set of data of the main analyzer over the whole explored Q range with a resolution of $\delta E$ = 3/1.5 meV, while white dots/squares are obtained by inspecting secondary analyzer (error bars as SD are within the size of the symbols). Shaded areas outline the energy range not experimentally accessible corresponding to the experimental resolution.

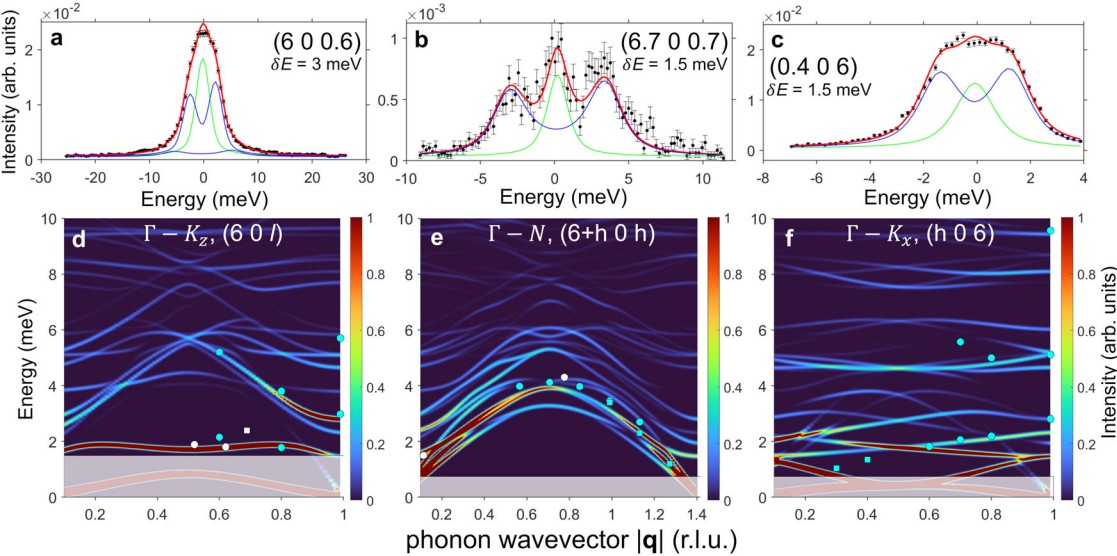

**Fig. 2 | Detecting very-low-energy phonon modes.** In **a**–**c** we show ID28 data on [VO(TPP)] (black scatters, with error bars representing the SE) obtained at $T = 300$ K with a resolution of $\delta E = 3$ meV for transverse scans along $\Gamma$–$K_z$ ($\mathbf{Q} = (6, 0, 0.6)$), and with $\delta E = 1.5$ meV along $\Gamma$–N ($\mathbf{Q} = (6.7, 0, 0.7)$) and $\Gamma$–$K_x$ ($\mathbf{Q} = (0.4, 0, 6)$) symmetry directions, respectively (with error bars representing the SE). Solid red lines are results of a fit obtained with ID28 custom software FIT28, comprising the excitation signals (blue lines) and the elastic one (green line). In **d**–**f** we show the IXS cross-section (colour map) simulated with pDFT phonon energies and polarization vectors along the same symmetry directions. Cyan dots/squares are experimental IXS excitation energies extracted from the complete set of data up to 10 meV of the main analyzer over the whole explored Q range with a resolution of $\delta E = 3/1.5$ meV, while white dots/squares are obtained by inspecting secondary analyzer (error bars as SD are within the size of the symbols). Shaded areas outline the energy range not experimentally accessible corresponding to the experimental resolution.

The data reported in Fig. 1h for a longitudinal scan along the $\Gamma$–$K_z$ direction show in fact that with this $\delta E$ we are able to detect modes with energies $\leq 2$ meV, here corresponding to a longitudinal acoustic and very-low-lying optical modes. Phonon modes with energies of the order of 3 meV are also present along the $\Gamma$–N direction, while transverse modes with energies $\leq 2$ meV are visible along the $\Gamma$–$K_x$ direction, corresponding to very-low-energy optical modes. Thus, IXS data on [VO(TPP)] show the presence of ultra-low-energy optical modes, clearly detectable with both low and high-resolution configurations. Inelastic spectra as a function of energy for other $\mathbf{Q}$ values are reported in Supplementary Figs. 2–7. As discussed below, all these findings are in excellent agreement with the simulations of the IXS cross-section obtained with DFT phonon energies and polarization vectors (see colour maps of Figs. 1, 2 and the following section).

## DFT simulation of IXS data

The unit cell of [VO(TPP)] is replicated three times along each crystallographic direction to obtain a $3 \times 3 \times 3$ supercell containing 4212 atoms. As described in the Methods section, the latter is optimized with pDFT and used to compute lattice force constants and phonons across the entire BZ. Phonon dispersions of [VO(TPP)] calculated with pDFT along the same symmetry directions in the reciprocal space explored experimentally are reported in Fig. 1c, f, i and Fig. 2d–f (see also Supplementary Fig. 8). The comparison with the phonon energies extracted from the IXS data, superimposed onto the same figures, demonstrates the optimal agreement with the experimental results. Importantly, pDFT calculations confirm the presence of the optical branches lying at very low energies along the $\Gamma$–$K_z$ and $\Gamma$–$K_x$ directions.

The evaluation of the spin-phonon couplings relies on a full description of phonon modes, comprising of both phonon energies $\omega_j(\mathbf{q})$ and polarization vectors $\boldsymbol{\sigma}_j^d(\mathbf{q})$. Therefore, the complete validation of the pDFT results requires the inspection of both these quantities before proceeding to the calculation of the spin dynamics. The phonon excitation intensities determined by the inelastic X-ray cross-section directly depend on phonon polarization vectors $\boldsymbol{\sigma}_j^d(\mathbf{q})$ (see Eq. (2) in Methods), and thus, IXS experiments also probe the composition

of phonon normal modes. In order to compare experimental data with DFT results, we simulated the IXS cross-sections starting from DFT-calculated phonon energies $\omega_j(\mathbf{q})$ and polarization vectors $\boldsymbol{\sigma}_j^d(\mathbf{q})$. The simulated cross-sections are reported as 2D colour maps in Fig. 1-c, f, i and Fig. 2d–f, where the colour code represents the excitation intensity. These maps provide immediate visualization of phonon dispersions and excitation intensities along the explored symmetry directions. However, for a more direct comparison between our simulations and IXS data, we inspected the cross-section as a function of energy for some representative $\mathbf{Q}$ values. From the results reported in Fig. 3, the excellent agreement between experimental and calculated IXS cross-section (using the experimental line-width) is evident for both low (panels a-e,g) and high-resolution data (panels f,h,i), thus validating both calculated phonon energies and polarization vectors. In particular, Fig. 3f–i highlight the contributions to the [VO(TPP)] IXS cross-section of the very-low-energy phonon modes. Only an overall rescaling of 10% has been uniformly applied to lower the phonon energies and better reproduce the IXS data. This correction is typical of pDFT calculations on molecular crystals[31] and, especially at low energy, is mainly due to van der Waals interactions corrections, as well as to temperature effects on the simulation cell[47]. It is also worth stressing that the calculated IXS excitation intensity of very-low-energy phonons strongly depends on the exact energy value through the phonon Bose factor $n_j(\mathbf{q})$. This high sensitivity of the IXS cross-section at very-low energy allowed us to address small discrepancies between data and calculations, of the order of tenth of meV, along $\Gamma$–$K_z$ and $\Gamma$–$K_x$ directions (see Fig. 3g, i). Comparisons bewteen inelastic spectra and calculated cross-sections for other $\mathbf{Q}$ values are reported in Supplementary Figs. 2–7.

## Spin-phonon coupling and decoherence simulation

The molecular qubit [VO(TPP)] contains a $^{51}V^{4+}$ ion, whose spin states can be described with the spin Hamiltonian

$$\hat{H}_S = \mu_B \mathbf{S} \cdot \mathbf{g} \cdot \mathbf{B} + \gamma_N \mathbf{I} \cdot \mathbf{B} + \mathbf{S} \cdot \mathbf{A} \cdot \mathbf{I}. \tag{1}$$

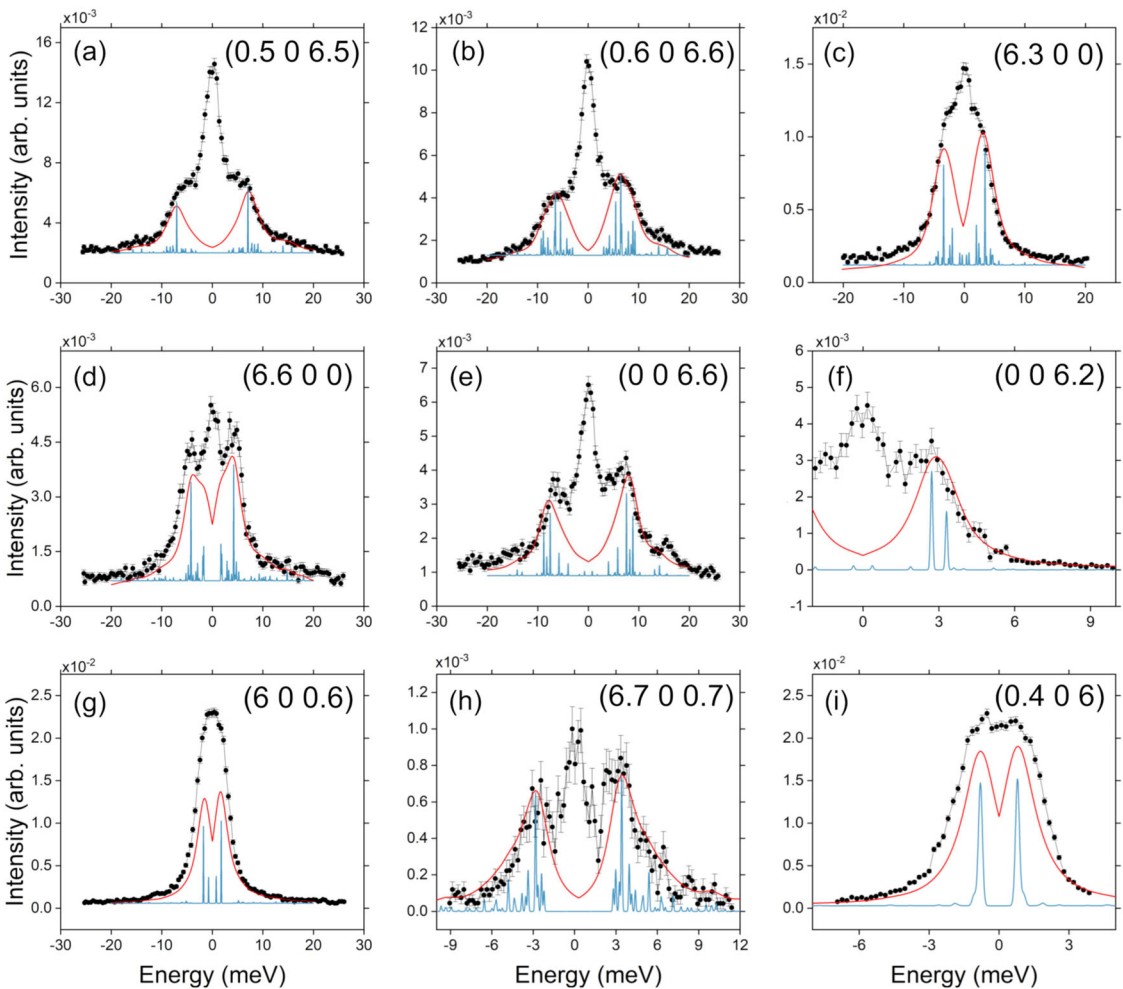

**Fig. 3 | Phononic excitations intensities: ID28 vs simulations.** ID28 data (black scatters, with error bars representing the SE)) along different symmetry directions and Q values compared with the simulated cross-section (red lines) calculated with the experimental resolution (**a**–**e**, **g**: $\delta E = 3$ meV; **f**, **h**, **i**: $\delta E = 1.5$ meV) and with a FWHM of 0.1 meV (blue lines) to distinguish the contribution of single phonon branches (elastic line omitted for clarity).

The first two terms in Eq. (1) describe the Zeeman interaction of the magnetic field with the electronic (**S**) and nuclear spin (**I**), respectively, where $\mu_B$ is the Bohr magneton, **g** is the Landé tensor, and $\gamma_N$ is the nuclear gyromagnetic factor. The third term, instead, corresponds to the hyperfine interaction (**A**) between the two spins. Coupling with other magnetic nuclei (e.g., $^1$H, $^{14}$N) is here neglected. The tensors **g** and **A** are computed with DFT using the pDFT-optimized structure and found in excellent agreement with experimental ones[27]. Results are reported in Table 1.

The simulation of spin-phonon relaxation with electronic structure methods requires the calculation of the effect of phonons on the Hamiltonian of Eq. (1). Here, we consider the modulation of the leading terms of Eq. (1), i.e. **g** and **A**. This corresponds to computing first- and second-order derivatives of these tensors with respect to phonon displacements, $q_{\alpha \mathbf{q}}$, i.e. $\mathbf{V}_{\alpha \mathbf{q}} = (\partial \mathbf{A}/\partial q_{\alpha \mathbf{q}})$, $\mathbf{V}_{\alpha \mathbf{q} \beta \mathbf{q}'} = (\partial^2 \mathbf{A}/\partial q_{\alpha \mathbf{q}} \partial q_{\beta \mathbf{q}'})$ and similarly for $\mathbf{g}^{26,38}$. The indexes $\alpha$ and $\mathbf{q}$ point to the phonon's band

index and reciprocal space vector, respectively. The total number of second-order derivatives scales quadratically with the number of molecular degrees of freedom and for [VO(TPP)], it would require a minimum number of $10^5$ DFT calculations. This volume of simulations is not sustainable with modern-day computational hardware and software. We solve this technical challenge by employing neural networks to efficiently interpolate DFT results[48]. The relation between the structure of [VO(TPP)] and the value of **A** and **g** is sampled 2000 times by applying random perturbations within the range $\pm 0.05$ Å to [VO(TPP)]'s optimized structure. Several neural networks with up to four hidden layers are trained on 1600 samples to predict the two tensors as a function of [VO(TPP)] atomic distortions. The prediction of the best-performing models is tested against DFT calculations for 200 samples not used at the training stage, revealing their excellent accuracy with a root mean squared error of 0.45 MHz and $8.2 \times 10^{-5}$ on the prediction of **A** and **g**, respectively (see Supplementary Figs. 9–11). The neural networks are then used to compute a 36-points numerical differentiation of the spin Hamiltonian coefficients with respect to all pairs of molecular degrees of freedom, for a total of $10^6$ evaluations of **A** and **g** (see Supplementary Fig. 12).

Figure 4 shows the comparison between the simulated phonon density of states and the norm of linear spin-phonon coupling coefficients, $|\mathbf{V}(\omega_\alpha)|$ (defined in the Methods section), for the **A** tensor as a function of the vibrations energy $\hbar \omega_{\alpha \mathbf{q}}$. Several important conclusions

**Table 1 | Simulated and experimental[27] spin Hamiltonian parameters for [VO(TPP)]**

|      | $g_\perp$ | $g_\parallel$ | $A_\perp$ (MHz) | $A_\parallel$ (MHz) |
|------|-----------|---------------|-----------------|---------------------|
| Exp. | 1.987     | 1.963         | 169             | 480                 |
| Sim. | 1.984     | 1.968         | 166             | 473                 |

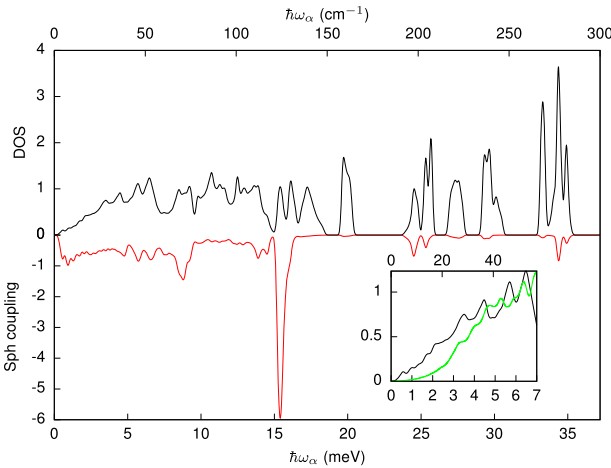

**Fig. 4 | Computed spin-phonon coupling and vibrational density of states.** The norm of the linear spin-phonon coupling coefficient as a function of the frequency of the vibration is reported in red along the negative axis in arbitrary units. The vibrational density of states is reported in black along the positive axis and in arbitrary units. The inset shows the comparison of the low-energy vibrational density of states for [VO(TPP)] (in black) and [VO(acac)$_2$][25] (green line).

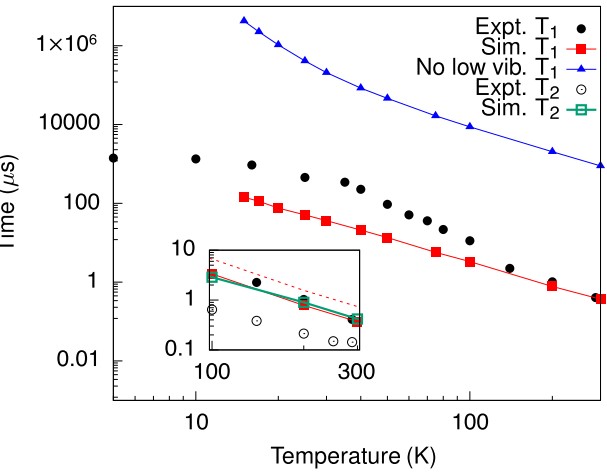

**Fig. 5 | Spin-phonon relaxation and decoherence time.** The computed spin-phonon relaxation time $T_1$ and coherence time $T_2$ (inset) are reported with a continuous red line and full squares, and a continuous green line and empty squares, respectively. The theoretical limit $T_2 = 2T_1$ is reported with a dashed red line in the inset. The simulated $T_1$ after removal of all phonons with $\hbar\omega_\alpha < 6$ meV is reported with a continuous blue line and filled triangles. Black full (empty) circles are used to report the inversion recovery (Hahn echo) experimental results of Ref. [27] (error bars are within the size of the symbols).

can be drawn from this result. Most importantly, this analysis shows a sizeable spin-phonon coupling for ultra-low-energy vibrations. In addition, the computed spin-phonon coupling norm and the phonons density of states show remarkably different profiles. This observation is in agreement with previous reports in other molecular complexes and stems from the fact that vibrational modes are varied in nature and depending on their symmetry[39] and localization on the first coordination shell[22,26,42], they will be more or less effective in coupling to the spin. Finally, the vibrational DOS does not follow the canonical Debye profile with ~$\omega^2$, but instead follows a linear dependency overlapped to a complex structure due to the many optical transitions falling at very low energy[25]. This demonstrates a breakdown of the simple Debye picture. The inset of Fig. 4 shows the comparison between the low-energy DOS of [VO(TPP)] and [VO(acac)$_2$]. Differently from the former, the latter shows a more typical Debye-like profile at low energy in virtue of the higher energy of optical phonons, with the first one computed at ~6 meV at the Γ-point[25,31].

According to recent literature[26,38], spin-phonon relaxation at temperature above ~10–20 K is due to a two-phonon Raman mechanism, where a spin transition is triggered by the absorption of one phonon and the simultaneous emission of a second phonon with similar energy. We simulate spin relaxation and decoherence due to this mechanism by solving the second-order secular Redfield equation including quadratic spin-phonon coupling terms $V_{\alpha q \beta q'}$[26,38] as implemented in the software MolForge[26] (see Supplementary Note 2 for more details on the workflow). As observed previously for vanadyl compounds[26,38], the simulations show the contribution of the modulation of the **A** tensor to be the leading relaxation mechanism in moderate magnetic fields such as 0.3 T employed in X-band EPR experiments. We note that the contributions of **A** and **g** are relatively close to one another in [VO(TPP)] for the values of the field considered and their contribution to spin-phonon relaxation follows qualitatively similar trends (see Supplementary Figs. 14–19).

Figure 5 reports the predictions of $T_1$ and $T_2$ in [VO(TPP)] against the experimental results obtained by inversion recovery and Hahn echo at X-band frequencies[27] (see plotted data points in Supplementary Tables 5, 6). We observe a good agreement between experimental and simulated $T_1$ time-scale above ~20 K and especially in the high-temperature regime, where simulations reveal a $T^{-2}$ profile, typical of two-phonon relaxation[26,38,49]. At lower temperatures, experiments are affected by cross-relaxation mediated by spin-spin dipolar interactions

and active due to the relatively high concentration (2%) of the V$^{4+}$ magnetic ions inside the isostructural diamagnetic Ti$^{4+}$ host[27].

The simulation of a high-temperature Raman profile ($T^{-2}$) from 10 K supports an interpretation of spin relaxation due to ultra-low-energy vibrations. We further test this hypothesis by manually removing the contribution to $T_1$ from all the vibrations with $\hbar\omega_\alpha < 6$ meV. These results are also reported in Fig. 5 and show a largely suppressed spin-phonon relaxation in these artificial conditions, thus confirming the relevance of low-energy phonons in limiting spin lifetime. This result is in agreement with the relevance of low-energy phonons in limiting spin lifetime, especially if they are strongly coupled to the spin, as shown for the investigated compound by our calculations. Since these phonons are always more populated than high-energy ones, they maintain a dominant role in the spin dynamics in any temperature condition.

Here we also report the DFT simulation of $T_2$ due to phonons and show that it follows the exact same temperature profile of $T_1$ and a trend $T_2 \leq T_1$. In particular, we simulate $T_2 \sim T_1$ for Raman relaxation due to the modulation of the hyperfine coupling (see inset of Fig. 5) and $T_2 \sim 0.8 T_1$ in the presence of the modulation of the g-tensor (see Supplementary Fig. 19). This finding is in stark contrast with the canonical relation $T_2 = 2T_1$ for phonon-limited spin coherence. We attribute this deviation to the presence of the pure dephasing contribution ($T_2^*$) to $T_2$, usually neglected for spin-phonon processes. As we show in Supplementary Note 4, $T_2^*$ vanishes for one-phonon processes, but becomes finite for two-phonon processes, like the ones considered here. The phonon-induced pure-dephasing decoherence mechanism has a simple physical interpretation. $T_2^*$ results from energy-conserving phonon processes, where a pair of degenerate phonons exchange energy among them, i.e. one phonon is absorbed by the spin and another one is simultaneously emitted. If the two phonons are not equally coupled to the spin, this process generates an effective magnetic noise at the spin site that leads to dephasing. Interestingly, there is an elegant parallel between this process and the one experienced by the spin coupled to a spin bath via dipole interactions. In the latter scenario, the cause for decoherence is energy-conserving flip-flop spin processes, where spins with an equal gyromagnetic factor exchange energy among them and cause dephasing of the central spin[50,51].

Importantly, our simulations of $T_2$ are in agreement with literature results that usually show $T_2 < T_1$[27,52–55] and reproduce the temperature

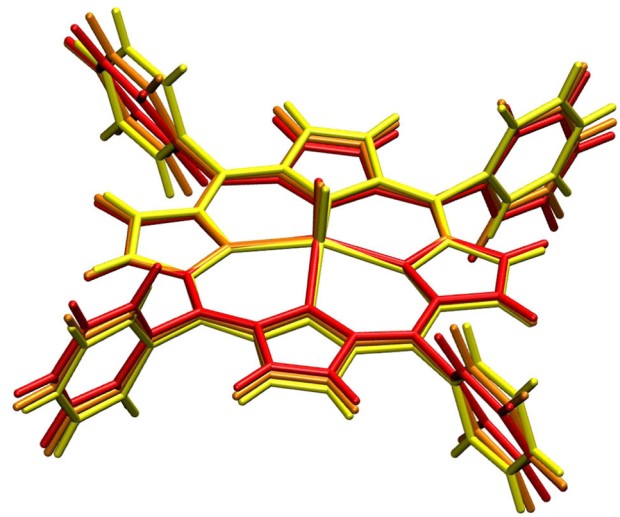

**Fig. 6 | [VO(TPP)] molecular distortions.** Two different degrees of molecular distortion associated with the first optical mode at the Γ-point are represented in red (large distortion) and orange (small distortion) colours. The yellow structure corresponds to the equilibrium molecular structure.

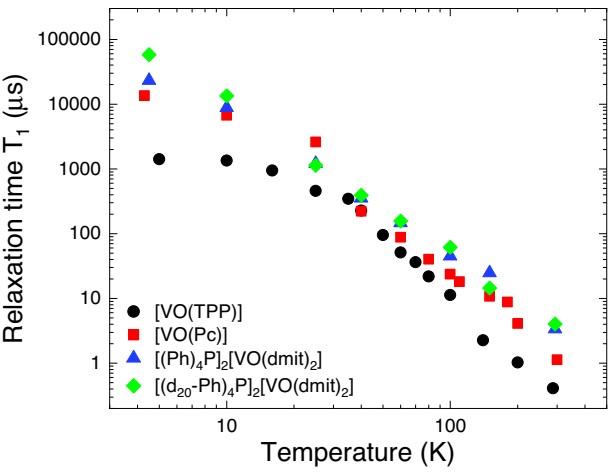

**Fig. 7 | Comparison with other VO-based systems.** [VO(TPP)] experimental spin-phonon relaxation time $T_1$ (black circles)[27] compared with other VO-based systems: VOPc (Pc = phthalocyanine, red squares)[53], [(Ph)$_4$P]$_2$[VO(dmit)$_2$] (dmit = 1,3-dithiole-2-thione-4,5-dithiolate, blue triangles)[54] and its analogue compound with a deuterated cation, [(d$_{20}$-Ph)$_4$P]$_2$[VO(dmit)$_2$] (green diamonds)[54]. All the samples were crystalline dispersions in their isostructural diamagnetic host: [TiO(TPP)] (2% dilution), TiOPc (0.1% dilution), [(Ph)$_4$P]$_2$[MoO(dmit)$_2$] (5% dilution) and [(d$_{20}$-Ph)$_4$P]$_2$[MoO(dmit)$_2$] (5% dilution) (error bars are within the size of the symbols).

dependence of the experimental decoherence time of [VO(TPP)] up to small rescaling factor. The discrepancy we observe in Fig. 5 is likely due to the here-neglected coupling of the electronic spin with other nuclei[56] (e.g., first coordination sphere $^{14}$N), contributing to the decoherence rate with additional pure dephasing terms with a similar temperature dependence. However, the ratio between $T_2$ and $T_1$ observed in the literature appears to be system-dependent. Intriguingly, a study on Nitrogen-Vacancy diamond defects has shown that $T_2 \sim 0.5 T_1$ after a dynamical decoupling pulse sequence was applied to entirely remove the contribution of spin-spin interactions and reveal the true phonon-limited $T_2$[57]. These results point to the urgent need to further investigate the relation between spin relaxation and decoherence to verify if there is a universal trend among $T_2$ and $T_1$.

In order to provide some chemical insight into the nature of the relevant phonons for spin relaxation, we study the molecular distortions associated with the first mode at the Γ-point, for [VO(TPP)] at -1.5 meV. As it can be appreciated from Fig. 6, this vibration corresponds to a large distortion of the phenyl rings overlapped to a bending of the porphyrin ring. Supplementary Movie 1 also shows that this molecular distortion is further combined with inter-molecular translations and rotations, where different molecules in the crystals change their reciprocal orientation and position. The visualization of these low-energy phonons thus shows that they are affected by both intra- and inter-molecular contributions.

## Discussion

Although previous experiments and simulations have shown that low-energy optical modes are a typical feature of MNMs[25,30–33,37,38], a full characterization of the role of low-energy optical phonons in the spin decoherence of a molecular qubit has never been provided before. In this work, we demonstrated the capabilities of our synergetic approach that combines IXS measurements and DFT spin dynamics simulations for a quantitative investigation of phonons in molecular qubits and their role in spin decoherence. To benchmark this approach we have chosen one of the most promising molecular qubits, the well-characterized [VO(TPP)] complex, being also one of the most radiation-robust of the vanadyl family. In particular, we have demonstrated the presence of very soft optical modes in [VO(TPP)], never seen before at such low energies, making this compound the ideal test-bed for investigating their contribution to spin decoherence. We have then provided a complete account of the contributions to spin

decoherence, including the pure dephasing induced by two-phonon Raman processes, a mechanism never discussed previously. Our analysis made it possible to uncover a wealth of additional insights on the spin dynamics of this molecular qubit that can hardly be extracted from the sole relaxation data. Indeed, the latter approach, based on a simple fitting of the Brons-Van-Vleck formula, failed to recognize the importance of low-energy vibrations and instead identified high-energy phonons as responsible for spin relaxation. Although it is not possible to exclude additional relaxation mechanisms taking place[26,58], the ones simulated here clearly represent an important contribution to $T_1$ and $T_2$ and they must therefore be addressed for improving coherence times.

Beyond evidencing the presence of the low-energy vibration modes that fully break down the Debye model in [VO(TPP)], we also identified their origin in the soft torsional degrees of freedom involving the rotation of the phenyl groups. These results point to the removal of the four phenyls rings from [VO(TPP)] (as in vanadyl porphyrin[52]) as a potential chemical strategy for tailoring the intra-molecular motions and thus slowing down relaxation. This conclusion is also supported by the fact that [VO(TPP)] is characterized by shorter spin-phonon relaxation times $T_1$ with respect to other VO-based systems[53,54] (see Fig. 7), where these phenyl groups are absent (for comparison with other VO-based systems in frozen solution[14] and decoherence times $T_2$ see Supplementary Figs. 22–23, respectively).

Since the majority of experimental studies on the relaxation dynamics of MNMs are performed in crystals or polycrystalline samples, the study of phonons in molecular crystals is the natural first step to understand spin relaxation and benchmark theoretical models. However, the long-term goal for applications is the embedding of single molecules in quantum devices. To check the persistence of the low-energy vibrations in single molecules and test the potential of the proposed chemical strategy, we therefore performed also gas-phase vibration calculations of [VO(TPP)] and of the vanadyl porphyrin (see Supplementary Movie 2 provided as Supplementary Material). Conversely, these simulations show that the gas-phase vibrations of [VO(TPP)] also exhibit low-energy modes with comparable frequency as the lattice's Γ-point. On the other hand, once the phenyl rings are removed, the lowest energy vibrational mode in the gas phase is shifted to 6.5 meV, a four-fold increase in frequency. This result points

to a potential increase in $T_1$ as the one reported in Fig. 5 after the removal of low-energy phonons. However, it should be stressed that this value represents an upper limit to what can possibly be achieved, as the presence of any condensed-matter environment will eventually reintroduce some low-energy vibrations due to the admixing of inter- and intra-molecular displacements.

In conclusion, our work revealed the vibrational contributions to spin decoherence in a prototypical molecular qubit, highlighting the importance of low-energy phonons for both spin relaxation and pure spin dephasing. Moreover, we have shown that ultra-low optical pho-nons in molecular crystals are likely originated by the presence of low-energy modes at the molecular level further shifted down in energy once combined with lattice vibrations, and that chemical engineering of molecular structures have the potential to considerably reduce the low-energy vibrational contributions and improve spin-relaxation time. The unprecedented insight into the nature of vibrational states, their coupling to spin, and their role in spin decoherence demonstrates that the IXS+DFT spin dynamics approach is the new technique of choice for investigating spin-phonon dynamics in mole-cular compounds and for the design of new systems.

## Methods

### Sample preparation

The [VO(TPP)] molecular qubit was synthesized according to the experimental procedure reported in ref. [27], and purified by thermal sublimation at 533 K (260°C) and $10^{-6}$ mbar for 48 h. After purification, single crystals suitable for IXS experiments (about 1 mm$^3$) were obtained by slow evaporation of a CH$_2$Cl$_2$/$n$-heptane (95:5) solution over two weeks.

### IXS experiment

The ID28 beamline at the European Syncrotron Facility is an inelastic X-rays spectrometer with energy and momentum transfer ranges particularly suited for studying phonons dispersions. The very small beam size (of the order of a few tens of $\mu$m) allows the investigation of systems available only in small quantities. The optical layout is based on the triple-axis principle, composed of the very high-energy reso-lution monochromator (first axis), the sample goniometer (second axis) and the crystal analyser (third axis). Thanks to its backscattering geometry and its length, ID28 is able to acquire a sufficient beam offset between the incident photon beam from the X-ray source and the very high-energy resolution beam focused at the sample position.

A single crystal of [VO(TPP)] with size of the order of $1 \times 1 \times 0.5$ mm$^3$ was mounted on ID28 side station in order to explore the reciprocal space with diffuse scattering and identify a suitable scat-tering plane. The sample was oriented in order to have ($h0l$) as the scattering plane (in conventional cell notation, see Supplementary Tables 1–3 and Supplementary Fig. 1) and explore the [VO(TPP)] reci-procal space along the $\Gamma$–N, $\Gamma$–K$_z$ and $\Gamma$–K$_x$ directions (($h0h$), ($00l$) and ($h00$) directions, respectively), in both longitudinal and transverse configurations. In particular, we focused on the 600 and 006 Bragg reflections. The sample was glued on a standard sample holder and placed on the ID28 sample stage, at the temperature of 300 K. We worked in transmission geometry and we exploited two different configurations of the ID28 silicon monochromator: Si(9 9 9), selecting an incoming energy of $E = 17.794$ keV with an energy resolution $\delta E = 3.0$ meV and Si(12 12 12) with an incoming energy of $E = 23.725$ keV and an energy resolution $\delta E = 1.5$ meV. The spectrometer layout, with 9 dif-ferent analysers within the same scattering plane (-0.75 deg spacing), enables the detection of 9 different momentum transfer simulta-neously. This characteristic of the ID28 instrument increases signi-ficantly the BZ sampling, obtained when the additional analysers aligns along a symmetry direction. Data on [VO(TPP)] were collected performing constant-Q energy scans for several Q values along the selected high-symmetry directions. Given the long tails of the

Lorentzian-like line-shape of the elastic signal (described with a damped-harmonic-oscillator function), we can only measure $|\mathbf{q}| \geq 0.1$ Å$^{-1}$, i.e., not to close to the selected Bragg peaks/$\Gamma$ points. Each dataset has also been normalized by the incident X-ray flux measured by the beam monitor before the sample.

### DFT calculations

The unit-cell X-ray structure of [VO(TPP)][27] was used as starting points for a periodic DFT optimization of a $3 \times 3 \times 3$ supercell with the soft-ware CP2K[59] (see Supplementary Table 4). Density functional theory (DFT) with the PBE functional[60], including Grimme's D3 van der Waals corrections[61], was used together with a double-zeta polarized (DZVP) MOLOPT basis set. A plane-wave cutoff of 1875 Ry was used. Lattice force constants and phonons were computed with the software MolForge[26]. Lattice force constants were computed with a two-point numerical differentiation of atomic forces with step of 0.01 Å. The tensors **A** and **g** were computed with the software ORCA[62] using DFT with the hybrid functional PBE0[63] and DKH-def2-SVP basis for C and H atoms and DKH-def2-TZVPP for all other atoms. All basis sets were decontracted and RIJCOSX was used as approximation for Coulomb and Hartree-Fock Exchange.

### Data analysis and simulations

IXS spectra as a function of energy were fitted by means of the custom beamline software FIT28 (see ref. [46]). The spectral components (elastic line and phonon excitations) were modelled with single-damped-harmonic-oscillator functions with the experimental resolution of each specific instrumental configuration. The Stokes and anti-Stokes intensities are corrected for the Bose–Einstein thermal population factor. It is worth stressing that the presence of more than one phonon mode within the experimental resolution with a non-zero cross-section results in an effective broadening of excitation lines, emerging from the summation of excitations close in energy.

The one-phonon IXS cross-section is defined as[45]

$$\frac{\partial^2 \sigma}{\partial \Omega \partial E} \propto \sum_{j,\mathbf{q}} \delta(\mathbf{Q} + \mathbf{q} - \mathbf{G}) |F_{j,\mathbf{q}}(\mathbf{Q})|^2$$
$$\frac{1}{\omega_j(\mathbf{q})} \left[ n_j(\mathbf{q}) \delta(\omega + \omega_j(\mathbf{q})) + (n_j(\mathbf{q}) + 1) \delta(\omega - \omega_j(\mathbf{q})) \right], \quad (2)$$

where $\omega_j(\mathbf{q})$ is the energy of the $j$th phonon branch. The momentum conservation law of the inelastic scattering events involve the exchanged scattering vector $\mathbf{Q}$, the phonon quasi-momentum $\mathbf{q}$ and the reciprocal lattice vector $\mathbf{G}$, while the energy conservation depends on the energy transfer $\omega$. $n_j(\mathbf{q}) = (\exp(\beta\omega_j(\mathbf{q})) - 1)^{-1}$ is the phonon Bose factor with $\beta = (K_B T)^{-1}$. The function $F_{j,\mathbf{q}}(\mathbf{Q})$ is the one-phonon structure factor, taking into account interference effects between the different atoms $d$ in the unit cell:

$$F_{j,\mathbf{q}}(\mathbf{Q}) = \sum_d \frac{f_d(\mathbf{Q})}{\sqrt{2m_d}} e^{i\mathbf{Q} \cdot \mathbf{R}_d} (\mathbf{Q} \cdot \boldsymbol{\sigma}_j^d(\mathbf{q})), \quad (3)$$

where $f_d(\mathbf{Q})$ is the X-ray atomic form factor, $m_d$ the mass and $\mathbf{R}_d$ the position vector in the real space of each atom, while $\boldsymbol{\sigma}_j^d(\mathbf{q})$ are the phonon polarization vectors.

Data simulations were performed by calculating the scattering cross-section in Eq. (2) with pDFT-calculated phonon energies $\omega_j(\mathbf{q})$ and normal modes polarization vectors $\boldsymbol{\sigma}_j^d(\mathbf{q})$. A rescaling of about 10% was uniformly applied to all the calculated phonon energies along all the symmetry directions. We then assumed a single-damped-harmonic-oscillator line-shape with the FWHM of the corresponding dataset (for the colour maps we used a FWHM = 0.1 meV, in order to distinguish the contribution of single phonon branches with a non-zero cross-section within the experimental resolution).

## Neural Networks

The neural networks used to interpolate spin Hamiltonian coefficients were built using the Keras API and Tensorflow library. The input layer contains 234 nodes, corresponding to 3N Cartesian coordinates with N being the number of atoms in the [VO(TPP)] molecule, and the output layer contains 9 nodes, corresponding to the tensor components of $\mathbf{A}$ or $\mathbf{g}$. The number of hidden layers and the number of nodes in each hidden layers are varied to obtain the model best-suited for $\mathbf{A}$ and $\mathbf{g}$. The sigmoid function is used as activation function for the hidden layers. The selected model for $\mathbf{A}$ has 2 hidden layers with 128 and 64 nodes in each hidden layer, while the model selected for $\mathbf{g}$ has 3 hidden layers with 128, 64, and 32 nodes in each hidden layer. The model obtained is trained with 1600 configurations of randomly distorted [VO(TPP)] molecule. The regularization hyperparameter is optimized for each model using a validation set of 158 configurations of [VO(TPP)] and the performance of the models is evaluated with a test set of 200 configurations that the models have never seen before (see Supplementary Note 2 and Supplementary Figs. 9–11 for further details).

## Spin-relaxation simulations

The trained machine learning models are used to calculate the first- and second-order derivatives, $\mathbf{V}_{\alpha \mathbf{q}} = (\partial \mathbf{A}/\partial q_{\alpha \mathbf{q}})$ and $\mathbf{V}_{\alpha \mathbf{q} \beta \mathbf{q}'} = (\partial^2 \mathbf{A}/\partial q_{\alpha \mathbf{q}} \partial q_{\beta \mathbf{q}'})$, numerically (Supplementary Fig. 12). The norm of the spin-phonon coupling coefficients for the modulation of $\mathbf{A}$ is defined as

$$|\mathbf{V}(\omega_\alpha)| = \frac{1}{N_q} \sum_{\mathbf{q}} \sum_{ij} \left| \frac{\partial A_{ij}}{\partial q_{\alpha \mathbf{q}}} \right|^2, \tag{4}$$

and similarly for the g-tensor. The coefficients are then used to model the spin relaxation with MolForge[26] (see Supplementary Note 3 for more details). The relaxation time $T_1$ is fitted from the $M_z$ dependence on time with a double exponential function (Supplementary Fig. 13a, c). The coherence time $T_2$ is fitted from the $M_\perp$ dependence on time with a double exponential function (Supplementary Fig. 13b, d). The external static magnetic field is set at 0.33 T along the $z$-direction. $T_1$ values were converged with respect to the number of q points used to sample the Brillouin zone and the size of the Gaussian smearing used to represent Dirac's delta function appearing in the Redfield equations[38,49] (Supplementary Figs. 14–20). A final mesh of $4 \times 4 \times 4$ q points and smear of 10 cm$^{-1}$ was used to obtain the temperature profile of $T_1$ and $T_2$.

## Data availability

Raw data from the IXS experiment were generated at the ESRF (proposal number HC-4312) and are available from the corresponding authors upon reasonable request. They will also be available on the ESRF Data Portal from 08/10/2023.

## Code availability

Matlab codes for data simulations are available from the corresponding authors upon reasonable request. The code MolForge was used to simulate phonons and spin dynamics, and it is available at "github.com/LunghiGroup/MolForge" and at https://doi.org/10.5281/zenodo.7596042.

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

## Acknowledgements

Dr. Yong Cai is gratefully acknowledged for useful discussions on IXS experiments. This project has received funding from the European Union's Horizon 2020 Research and Innovation Programme under grant agreement No 862893 (FET-OPEN project FATMOLS) (S.C.) and from the European Research Council (ERC) (grant agreement No. [948493]) (A.L.). It was also supported by the Italian MIUR with the Progetto Dipartimenti di Eccellenza 2018-2022 (ref. B96C1700020008) (F.T., R.S.), by Fondazione Cariparma (S.C.) and The National Recovery and Resilience Plan, Mission 4 Component 2 - Investment 1.4 - NATIONAL CENTER FOR HPC, BIG DATA AND QUANTUM COMPUTING—funded by the European Union—NextGeneration EU - CUP B83C22002830001 (F.T.). We also acknowledge the European Synchrotron Radiation Source for instrument time on the ID28 beamline (proposal number HC-4312) (E.G.) and the Trinity College Research IT and the Irish Centre for High-End Computing (ICHEC) for computational resources.

## Author contributions

E.G., P.S., R.S. and S.C. proposed the use of inelastic X-ray scattering to measure phonons in molecular qubits. E.G., S.Ch. and L.P. performed the experiment after discussion with C.M. and R.C. on a single crystal

sample synthesized by F.S. Data treatment was made by S.Ch. and L.P., while A.A., F.T. and A.L. performed DFT calculations. Data analysis and simulations were made by E.G., S.Ch. and S.C. A.L. developed the neural-network approach and performed spin dynamics simulations with V.H.A.N. E.G., A.L. and S.C. wrote the manuscript with inputs from all coauthors.

## Competing interests

The authors declare no competing interests.
