## [Peer Review File · Nature Communications]

REVIEWER COMMENTS

Reviewer #1 (Remarks to the Author):

This manuscript applies computation and experimental spectroscopy to the question of what molecular factors are influencing spin relaxation in a V(IV) complex. The authors apply inelastic X-ray scattering to analyse the low-energy spectrum of the VO(TPP) molecule and combine that data with ab initio studies to explore the exact vibrations that could be driving spin relaxation. Ultimately two conclusions are reached: (1) that ultra-low optical phonons are a combination of molecular vibrations that are shifted down in energy by mixing with lattice vibrations and (2) that chemical engineering of structure will allow the synthetic chemist to improve spin-lattice relaxation by limiting the low-energy vibrational contributions. I think the work is well done, but I do not think the paper meets the impact standards that should be in place for Nature Comm based on the below comments. I want to stress that I do like the paper and would be happy to see it published elsewhere, I just think it is a more specialized work at this point. Strengthening the manuscript in response to the below points will help broaden its appeal.

The main conclusion of the paper (#2 above) is important and relevant to the field, but I do not think it is particularly new. Many experimental and computational papers have demonstrated that molecular finetuning will control spin-lattice relaxation (J. Am. Chem. Soc. (2021), 143(42), 17305-17315; Chem. Sci. (2022), 13, 7034; Inorg. Chem. (2018), 57, 731-740). It also is not new in terms of the insight that limiting the low-energy modes is advantageous (Inorg. Chem. (2021), 60, 14096-14104). It is new that the experiments and computations appear to trace the effect to a specific type of vibration, but without better comparison with other complexes, the study comes across as a single point, when a more convincing analysis would bring in many other complexes for evidence.

Finally, this is maybe the fifth paper I've read recently about the same VO(TPP) complex or very close analogues (Chem. Sci. (2021), 12(36), 12046-12055; J. Am. Chem. Soc. (2018), 140(38), 12090-12101; J. Phys. Chem. A. (2020), 124(44), 9252-9260; J. Am. Chem. Soc. (2021), 143(42), 17305-17315). It is not totally clear to me why this complex in particular is being focused on so much, and I think the field is getting stuck here instead of pushing the envelope. Examples of new molecules would have strengthened the paper in terms of novelty, or perhaps more examples (related to the paragraph above) to truly demonstrate a widely applicable design principle. That would be really exciting! I do understand why the molecule was chosen here because of radiation stability, according to the authors. A bit more detail on what the authors mean by this would make the motivation for the molecule stronger to the reader.

Reviewer #3 (Remarks to the Author):

The present manuscript prepared by E. Garlatti et al. investigated the importance of low energy lattice phonon in the relaxation-process of a molecular spin qubit candidate [VO(TPP)] by means of inelastic X-ray scattering (IXS) technique and ab initio spin dynamics. Using IXS, both acoustic and optical phonon branches are obtained. Further, the Redfield relaxation theory is applied to determine longitudinal relaxation time (T_1), where the spin-phonon couplings coefficient is obtained by a kind of machine learning technique (neural network-based interpolation) on ab initio results. The authors found that ultra-low energy (8 to 16 cm^{-1}) optical phonon with intra-molecular nature plays a critical role in determining the relaxation time of the molecular spin qubit.

The spin-vibration coupling is a hot topic in molecular nanomagnets and molecular spin qubits, while the role of ultra-low energy vibrations is not clear. Overall, the present work is solid, and manuscript is well written also the theoretical investigation is in good agreement with experimental observation. Thus, I support the publication of the present manuscript in nature communication after author address following comments.

1- The authors investigated T_1 in this complex, while they didn't consider transverse relaxation time (T_2). In fact, T_2 is the most important feature for molecular spin qubits. Can the authors compute T_1 contribution to T_2 to increase the impact of their study, as T_1 is extensively studied by them in ref [24-28]. As the authors are using full Redfield theory (eq. S1), they can prepare their density from superposition state of their eigen spin Hamiltonian.

2- My next concern is regarding spectral function, eq. S3. This includes no anharmonic term. Is there any specific reason for that? The problem is that one could use any value of σ (smearing effect) to fit their experimental T_1 curve.

3- Lastly, the authors show in Fig 5 that by excluding low energy phonons, one could increase the relaxation time of molecular spin-qubit substantially. Can the authors comment on how to achieve this exclusion of these low energy phonons from synthetic chemistry point of view?

Report on

“The critical role of ultra-low energy vibrations in the relaxation dynamics of molecular qubits”

By E.Garlatti et al.

The authors report inelastic X-ray scattering experiments on one air stable oxo-vanadium complex which was first reported and characterized in a previous work (J.Am.Chem.Soc. 2018,140, 12090-12101) as very promising qbit type complex entering as a building motive in an metal-organic-framework. In the present work, the authors demonstrate the power of the X-ray scattering spectroscopy and its advantages over the inelastic neutron scattering spectroscopy allowing one to overcome the need of isotopic substitution of hydrogen with deuterium and allowing to use millimeter size single crystals. The authors demonstrate convincingly the importance of intra-molecular lattice vibrations of very low energy (1-2 meV) on the spin relaxation time at ambient temperatures. The work finds strong support by theoretical calculations of phonon dispersions and spin-phonon coupling matrix elements of the spin-Hamiltonian parameters showing the power of first principle methods like DFT, machine learning and neural networks to treat periodic structures with qbit type complexes. In this sense, I find the present work a valuable addition to the field of molecular magnetism and recommend publication in Nature Communication.

However, the reviewer encountered problems with the presentation of the results and found some inconsistencies and errors.

1. The structure of VO(tpp) was reported previously (J.Am.Chem.Soc. 2018,140, 12090-12101) and showed a disorder of oxygen which certainly is also present in the sample they measured. How does this disorder affect computed phonon energies and spin-lattice relaxation times?
2. Table S1 in the ESI lists cell parameters for the TiOtp (a=b=13.3796 Å,c=9.7766 Å), rather than for VOtp (a=b=16.644 Å,c=13.856 Å). In the J.Am.Chem.Soc. 2018,140, 12090-12101 paper, it was stated, that being isostructural, the coordination geometries of TiOtp and VOtp are similar, however the comparison shows that these differ significantly (V: V-O 1.696 Å, V-N 2.149 Å, \angle 111.80°, deviation of V from the N₄ plane 0.798 Å; Ti: Ti-O 1.630 Å, V-N 2.104 Å, \angle 105.42°, deviation of Ti from the N₄ plane 0.559 Å), so what was the starting geometry taken in the periodic DFT calculations? Could the authors please list periodic DFT optimized geometries, lattice parameters and fractional coordinates and compare them with X-ray geometry reported in the ja8b06733_si_002.cif file.
3. The authors have chosen the N, Kx and Kz symmetry points in the Brillouin zone; wave vector symmetries at these points would greatly simplify the analysis; these are worth to include in Table S2.
4. In the Methods section there is no mention about the preparation and the orientation of the 1x1x0.5 mm³ single crystal they used.

- There is an error in Table I: entries 480 and 473 belong to $A_{||}$, entries 169 and 166 belong to A_{\perp} .
- The reviewer has tried to reproduce “Exp” and “Sim(DFT?)” spin Hamiltonian parameters reported in Table I (PBE0 functional). Calculations of \mathbf{g} - and \mathbf{A} -matrices is always tricky because of the ambiguity in choosing a proper functional; benchmarking results against values of the spin Hamiltonian parameters is an important preliminary step of every analysis of this kind. There is a thorough study on this, see Mol.Phys.105,2007, 2049-2071. The reviewer used the PBE geometry of the molecular VOtp complex (V: V-O 1.560 Å, V-N 2.084 Å, $\angle 104.30^\circ$, deviation of V from the N_4 plane 0.514 Å). Results are listed in the Table below. As pointed out in the cited reference hybrid functionals are superior to meta-GGA. However, parameters computed using PBE0 (as stated in the Methods section “DFT calculations”) differ from the “Sim” values in Table I; is it possible the B3LYP rather than PBE0 functional was used in the simulations?

DFT functional	g_{\perp}	$g_{ }$	$ A_{\perp} $ (MHz)	$ A_{ } $ (MHz)
BP86	1.990	1.978	93	212
PBE	1.990	1.978	88	217
PBE0	1.987	1.971	207	526
B3LYP	1.987	1.971	164	488
TPSSh	1.990	1.978	173	481
B2PLYP	1.976	1.961	246	588
“Exp”	1.987	1.963	169	480

- A work-flow diagram of the theoretical simulations would be very helpful for those who are interested to apply the computational protocol. I can imagine (if correct...) that: a) a periodic DFT geometry optimization followed by Hessian calculation at the Γ point of the Brillouin zone is first carried out (CP2K). Calculated frequencies are worth comparing with similar calculations of the VOtp complex in the gas phase. b) The force field is passed to MolForge to compute the phonon spectrum; c) 1600 randomly distorted configurations are passed to the ORCA package providing a set of \mathbf{g} and \mathbf{A} matrices (python script?). d) The data is passed to the neural network algorithm to interpolate and provide first and second derivatives with respect to the phonon modes $q\alpha$ (python script?) e) spin-phonon coupling matrix elements are passed to the algorithm given by eq.S1-S4 (notations not specified, say α, β ?, $V_{db}^{\alpha-q\beta-q'}$?, there may be some misprints in eqs S2 and S3?) to compute spin-lattice relaxation times (MolForge).
- Figure 4 reports spin-phonon coupling and vibrational density of states. On Figure 7(JACS, 2018,140, 12090 a vibrational transition at 67 cm^{-1} was detected in the THz spectra (8.3 meV) which is difficult to identify in the black trace showing the phonon density on Figure 4. The same transition was postulated to affect the relaxation time (eq.7) and second higher frequency vibration 303 cm^{-1} was deduced from the temperature dependence of the relaxation time. It is highly recommended to harmonize the results of the two studies. In particular modelling the relaxation times based on the extended Brons-van Vleck model should be compared with the present more advanced treatment of spin-phonon coupling.
- It is the opinion of the present reviewer, that the theory part may be improved to reach crucial points of chemistry, i.e. to improve spin-lifetimes by rational design based on spectroscopy and theory. For example, it is not clear why ultra-low energy vibrations affect the relaxation times up to room temperatures. Presumably such modes have a big vibrational overlap with local

vibrations which can be quantified by van-Vleck coefficients $a_{i,q\alpha}$ (Phys.Rev.1940,57,426), i.e. the projections of phonons q_α on the local modes Q_i pertaining to a VOtpp unit:

$Q_i = \sum_{q\alpha} a_{i,q\alpha} q_\alpha$ ($i = 1 : 228$) (see also J.Phys.Chem.Lett. 2017, 8, 1695), and corresponding

expression for spin-phonon matrix elements; $(d\mathbf{g} / dQ_i)_o = \sum_{q\alpha} (d\mathbf{g} / dq_\alpha)_o q_\alpha$ ect.

10. Throughout the manuscript one refers to the theoretical calculations as “ab initio” which is preserved for wave-functions based methods, so may be one should better use “DFT”, or “Kohn-Sham DFT”.

We thank the three Reviewers for the careful reading of our manuscript. We believe that their indications and criticisms helped us in significantly improving our work. We thank **Reviewers 2 and 3 for recognizing the importance of our results and for recommending publication in Nature Communications** and **Reviewer 1 for stating that “the work is well done” and that “I do like the paper”**.

We believe that some weaknesses in the presentation of the experimental results and the discussion of their implications could have led Reviewer 1 to raise concerns about the impact of our work. We have modified the paper to make the importance and novelty of our results more evident. This is in fact the **first application of inelastic X-ray scattering (IXS) to molecular nanomagnets**. Our results demonstrate that the present **combination of IXS with DFT calculations is the new technique of choice** to investigate the phonon-induced relaxation dynamics of molecular qubits.

We also addressed all the remarks of Reviewers 2 and 3. In particular, our manuscript now reports **the first DFT calculation of the spin coherence time T_2 in molecular nanomagnets including the crucial pure-dephasing contribution due to the two-phonon Raman processes**.

Please find below the detailed answers to all the remarks and the corresponding modifications of the manuscript.

Response to Reviewer #1

“This manuscript applies computation and experimental spectroscopy to the question of what molecular factors are influencing spin relaxation in a V(IV) complex. The authors apply inelastic X-ray scattering to analyse the low-energy spectrum of the VO(TPP) molecule and combine that data with ab initio studies to explore the exact vibrations that could be driving spin relaxation. Ultimately two conclusions are reached: (1) that ultra-low optical phonons are a combination of molecular vibrations that are shifted down in energy by mixing with lattice vibrations and (2) that chemical engineering of structure will allow the synthetic chemist to improve spin-lattice relaxation by limiting the low-energy vibrational contributions. I think the work is well done, but I do not think the paper meets the impact standards that should be in place for Nature Comm based on the below comments. I want to stress that I do like the paper and would be happy to see it published elsewhere, I just think it is a more specialized work at this point. Strengthening the manuscript in response to the below points will help broaden its appeal.”

We thank the Referee for stating that “the work is well done” and for making evident that a major consequence of our results was not clear in the previous version of the paper. Indeed, our aim is to demonstrate that IXS has the sensitivity and the power to become the new standard approach to investigate phonons and vibrations in molecular qubits and in molecular nanomagnets in general. This will have a large impact on future experimental investigations in molecular magnetism, but it was not highlighted enough, leading the Reviewer to conclude “I do not think the paper meets the impact standards”.

We have now significantly rewritten the introduction and discussion to make this point and its large impact much more evident.

Moreover, we now added the first ever DFT calculation of phonon-induced pure dephasing in molecular nanomagnets (see detailed Response to Reviewer 3).

“The main conclusion of the paper (#2 above) is important and relevant to the field, but I do not think it is particularly new. Many experimental and computational papers have demonstrated that molecular finetuning will control spin-lattice relaxation (J. Am. Chem. Soc. (2021), 143(42), 17305-17315; Chem. Sci. (2022), 13, 7034; Inorg. Chem. (2018), 57, 731-740). It also is not new in terms of the insight that limiting the low-energy modes is advantageous (Inorg. Chem. (2021), 60, 14096-14104). It is new that the experiments and computations appear to trace the effect to a specific type of vibration, but without better comparison with other complexes, the study comes across as a single point, when a more convincing analysis would bring in many other complexes for evidence.”

This important comment demonstrates again that one of our main messages was not clear, the first demonstration of the power of the combined IXS + ab-initio approach. Indeed, here we present the first measurements of phonons in a molecular qubit (and in molecular nanomagnets in general) obtained with IXS. This technique has in fact several advantages with respect to other spectroscopy techniques for the investigation of phonons and molecular vibrations such as inelastic neutron scattering, crucially the possibility to investigate very small single crystals, the energy-independent resolution and the very small background.

Moreover, we found ultra-low energy optical phonon modes at about 1-2 meV, a record that also outdoes other ordered systems with low-energy optical phonons, like thermoelectric materials. These modes play a strong role in relaxation as shown by our ab-initio calculations. It is important to note that IXS directly probes phonon eigenvectors and not only eigenvalues. Hence, the conclusions drawn in the present work are based on an extremely solid ground. This study goes much beyond the other quoted by the Referee on this. For instance, in Inorg. Chem. (2021), 60, 14096-14104 the authors exploit infrared spectroscopy to probe excitations in the examined compound. This technique not only is limited to the investigation of polar optical modes at $q \approx 0$, but it doesn't even have the same resolution of the here-proposed IXS technique (1.5 meV), preventing also to investigate modes at very-low energies (below 3 meV), which are crucial for magnetic relaxation.

Also from the theoretical point of view, this work represents the second simulations of spin relaxation in $S=1/2$ with the inclusion of out-of-Gamma phonons. Only thanks to this enormous computational effort (simulation of 4,500 atoms with pDFT) we were able to shine light on the nature and importance of ultra-low-energy vibrations. In addition, this is the first time that EPR inversion recovery experiments on diluted crystals are compared with this type of simulations, providing a first of its kind benchmark for simulations and the best reproduced experimental results yet. Now that we have extended our study to coherence time (showing that the usually neglected pure dephasing arising from Raman process is actually important), our work includes fundamental new results that will be of importance for the entire community interested in spin relaxation, quantum technology, sensing, magnetic resonance and more.

We have now significantly modified the paper to make these points clearer and we moved the comparison with other similar compounds from the Supplementary Information to the main text.

“Finally, this is maybe the fifth paper I’ve read recently about the same VO(TPP) complex or very close analogues (Chem. Sci. (2021), 12(36), 12046-12055; J. Am. Chem. Soc. (2018), 140(38), 12090-12101; J. Phys. Chem. A. (2020), 124(44), 9252-9260; J. Am. Chem. Soc. (2021), 143(42), 17305-17315). It is not totally clear to me why this complex in particular is being focused on so much, and I think the field is getting stuck here instead of pushing the envelope. Examples of new molecules would have strengthened the paper in terms of novelty, or perhaps more examples (related to the paragraph above) to truly demonstrate a widely applicable design principle. That would be really exciting! I do understand why the molecule was chosen here because of radiation stability, according to the authors. A bit more detail on what the authors mean by this would make the motivation for the molecule stronger to the reader.”

IXS is a never-before-used technique in the field of molecular magnetism, thus to demonstrate its power requires to start from a sound benchmark and we chose one of the most investigated molecular qubits. Having a deep knowledge of our [VO(TPP)] single crystal sample (radiation robustness, crystal properties and indexing, energy range of magnetic transitions,..) was crucial from an experimental point view and contributed a lot to the success of such difficult measurements. Moreover, [VO(TPP)] is very interesting because it matches several requirements for its exploitation as a qubit in quantum technologies. For instance, its electronic spin 1/2 is coupled to the ⁵¹V nuclear spin 7/2 via hyperfine interactions, yielding a qubit–qudit unit, that can be coherently controlled via suitable pulse sequences to implement quantum error correction and quantum simulation algorithms. Both nuclear and electronic coherence times are much longer than the time required to manipulate the system. Furthermore, the porphyrin ligands of [VO(TPP)] are also a resource for the scalability of molecular qubits, allowing the coupling of multiple and distinguishable centers (see, for instance, 10.26434/chemrxiv-2022-1v5b4 and 10.26434/chemrxiv-2022-wx7lc). For all these reasons, [VO(TPP)] and other “very close analogues” are really very promising candidates for molecule-based quantum technologies, and a deep understanding of both their coherent and incoherent dynamics is therefore required.

We would also like to stress that IXS experiments are inherently time consuming, since they require a peer-review procedure of the proposed experiment and sample(s) to obtain beamtime at the large-scale facility of choice. In addition, a typical IXS experiment on phonons requires several shifts of measurements to be able to measure enough points in the reciprocal space with good statistics. Thus, it is very hard to extend the experimental analysis reported in this work to a series of compounds in a single shot.

However, we agree with the Referee that comparing these findings with other compounds is extremely useful. Hence, in the last part of the paper we compare the behavior of [VO(TPP)] with those of similar compounds, finding the relaxation time of [VO(TPP)] shorter than the others as expected. This comparison and the associated discussion further demonstrate the validity of our conclusions on the design principle.

We have modified the paper to underline these points and we moved the comparison with other similar compounds to the main text.

Response to Reviewer #2

“The authors report inelastic X-ray scattering experiments on one air stable oxo-vanadium complex which was first reported and characterized in a previous work (J.Am.Chem.Soc. 2018,140, 12090-12101) as very promising qbit type complex entering as a building motive in an metal-organic-framework. In the present work, the authors demonstrate the power of the X-ray scattering spectroscopy and its advantages over the inelastic neutron scattering spectroscopy allowing one to overcome the need of isotopic substitution of hydrogen with deuterium and allowing to use millimeter size single crystals. The authors demonstrate convincingly the importance of intra-molecular lattice vibrations of very low energy (1-2 meV) on the spin relaxation time at ambient temperatures. The work finds strong support by theoretical calculations of phonon dispersions and spin-phonon coupling matrix elements of the spin- Hamiltonian parameters showing the power of first principle methods like DFT, machine learning and neural networks to treat periodic structures with qbit type complexes. In this sense, I find the present work a valuable addition to the field of molecular magnetism and recommend publication in Nature Communication.”

We gratefully thank the reviewer for appreciating our work and for fully supporting its publication in Nature Communication.

“The structure of VO(tpp) was reported previously (J.Am.Chem.Soc. 2018,140, 12090-12101) and showed a disorder of oxygen which certainly is also present in the sample they measured. How does this disorder affect computed phonon energies and spin-lattice relaxation times?”

We optimized the unit-cell structure of both limiting cases, i.e. when there is an inverse centre and when there is not. We found that DFT predicts the cell with the inverse center to be more stable than the other and therefore we utilized the former structure for all subsequent simulations. Accounting for disorder explicitly in simulations is of course not possible, as it would require using very large supercells and giving up translational invariance, which is crucial for integrating the Brillouin zone. However, we would like to stress that the phonons are probably very little affected by the VO bond disorder, given the fact that this chemical group does not interact strongly with neighboring molecules. Inter-molecular interactions in the [VO(TPP)] crystal are expected to be dominated by the vdW contribution of aromatic groups. This is ultimately validated by the fact that we can accurately reproduce experimental phonons both in terms of energies and normal modes.

“Table S1 in the ESI lists cell parameters for the TiOtp (a=b=13.3796 Å, c=9.7766 Å), rather than for VOtp (a=b=16.644 Å, c=13.856 Å). In the J.Am.Chem.Soc. 2018,140, 12090-12101 paper, it was stated, that being isostructural, the coordination geometries of TiOtp and VOtp are similar, however the comparison shows that these differ significantly (V: V-O 1.696 Å, V-N 2.149 Å, 111.80°, deviation of V from the N4 plane 0.798 Å; Ti: Ti-O 1.630 Å, V-N 2.104 Å, 105.42°, deviation of Ti from the N4 plane 0.559 Å), so what was the starting geometry taken in the periodic DFT calculations? Could the authors please list periodic DFT optimized geometries, lattice parameters and fractional coordinates and compare them with X-ray geometry reported in the ja8b06733_si_002.cif file.”

The X-ray geometry reported in the ja8b06733_si_002.cif file as a supporting information of the paper by Yamabayashi, et al. (J.Am.Chem.Soc. 2018,140, 12090-12101) is that of the [VO(TCPP-Zn₂-bpy)] complex, i.e., the vanadyl-based 3D MOF. The .cif file for the two isostructural compounds [TiO(TPP)] and [(VO)TPP] is ja8b06733_si_003.cif. The cell parameters reported in this file are the same as in our Table S1. This [VO(TPP)] crystal structure was used as the starting point of our computational

investigation. We have now included in ESI the optimized lattice parameters and the optimized geometry of the molecule later used for the ab initio and spin dynamics simulations.

“The authors have chosen the N, Kx and Kz symmetry points in the Brillouin zone; wave vector symmetries at these points would greatly simplify the analysis; these are worth to include in Table S2”

The definition of the Brillouin Zone symmetry points is reported in the Supplementary Information in Table S3.

“In the Methods section there is no mention about the preparation and the orientation of the 1x1x0.5 mm³ single crystal they used.”

We have now included a “Sample preparation” paragraph in the Methods section of the manuscript and we have more clearly explained the orientation of the crystal in the description of the IXS experiment.

“There is an error in Table I: entries 480 and 473 belong to $A_{||}$, entries 169 and 166 belong to A_{perp} ”

We thank the reviewer for spotting out this typo, which we have now corrected.

“The reviewer has tried to reproduce “Exp” and “Sim(DFT?)” spin Hamiltonian parameters reported in Table I(PBE0 functional). ...”

We have now performed a full screening of A and g with the DFT functionals used by the reviewer and realized that their results were obtained without decontracting the basis set. As we noted in the Methods sections, we employed a fully decontracted basis to improve the description of core states, which are crucial to the determination of accurate hyperfine constants.

DFT Functional	g_{\perp}	$g_{ }$	A_{\perp} (MHz)	$A_{ }$ (MHz)
PBE0 decontracted basis	1.984	1.968	166	473
PBE0 non-decontracted basis	1.984	1.968	221	533
B3LYP decontracted basis	1.985	1.969	128	443
B3LYP non-decontracted basis	1.985	1.970	174	493

“A work-flow diagram of the theoretical simulations would be very helpful for those who are interested to apply the computational protocol.”

We thank the reviewer for their suggestion. We have now expanded Supplementary Note 3 to include a detailed work-flow of spin-dynamics simulations.

“Figure 4 reports spin-phonon coupling and vibrational density of states. On Figure 7(JACS, 2018,140, 12090 a vibrational transition at 67 cm⁻¹ was detected in the THz spectra (8.3 meV) which is difficult to identify in the black trace showing the phonon density on Figure 4. The same transition was postulated to affect the relaxation time (eq.7) and second higher frequency vibration 303 cm⁻¹ was deduced from the temperature dependence of the relaxation time. It is highly recommended to harmonize the results of the two studies. In particular modelling the relaxation times based on the extended Brons-van Vleck model should be compared with the present more advanced treatment of spin-phonon coupling”

As the reviewer correctly noted, the previous study did not avail of an advanced computational methods as the one presented here. Highlighting the importance of providing such a detailed picture of spin-phonon relaxation is a key message of our work and we have therefore introduced a new paragraph in the discussion section where we discuss this point. In short, we believe that fitting experimental data with the Brons Van-Vleck relation is prone to overfitting and that a combined theoretical and experimental approach is to be preferred.

We would also like to reassure the reviewer that simulations are consistent also with respect to the previous experimental work, where a THz spectrum was recorded. We now added a Gamma-point simulation of the IR spectrum of [VO(TPP)] to Supplementary Note 3, which nicely reproduces the features around 70 cm⁻¹.

“It is the opinion of the present reviewer, that the theory part may be improved to reach crucial points of chemistry, i.e. to improve spin-lifetimes by rational design based on spectroscopy and theory. For example, it is not clear why ultra-low energy vibrations affect the relaxation times up to room temperatures.”

Low energy phonons are able to affect relaxation up to ambient temperature simply because their population will always be larger than any other high-energy mode and also are optical in nature, as depicted in Fig. 6, and are therefore able to strongly couple to spin. The combination of these two features makes them the dominant relaxation channel at any temperature. The only way for high-energy modes to dominate Raman spin relaxation would be for them to be coupled to spins orders of magnitude more strongly than what they are.

We added a comment on this in the paper.

“Throughout the manuscript one refers to the theoretical calculations as “ab initio” which is preserved for wave-functions based methods, so may be one should better use “DFT”, or “Kohn-Sham DFT”.

We have revised the text to account for this.

Response to Reviewer #3

“The present manuscript prepared by E. Garlatti et al. investigated the importance of low energy lattice phonon in the relaxation-process of a molecular spin qubit candidate [VO(TPP)] by means of inelastic X-ray scattering (IXS) technique and ab initio spin dynamics. Using IXS, both acoustic and optical phonon branches are obtained. Further, the Redfield relaxation theory is applied to determine longitudinal relaxation time (T_1), where the spin-phonon couplings coefficient is obtained by a kind of machine learning technique (neural network-based interpolation) on ab initio results. The authors found that ultra-low energy (8 to 16 cm^{-1}) optical phonon with intra-molecular nature plays a critical role in determining the relaxation time of the molecular spin qubit.

The spin-vibration coupling is a hot topic in molecular nanomagnets and molecular spin qubits, while the role of ultra-low energy vibrations is not clear. Overall, the present work is solid, and manuscript is well written also the theoretical investigation is in good agreement with experimental observation. Thus, I support the publication of the present manuscript in nature communication after author address following comments.”

We gratefully thank the reviewer for appreciating our work and for fully supporting its publication in Nature Communications.

“The authors investigated T_1 in this complex, while they didn’t consider transverse relaxation time (T_2). In fact, T_2 is the most important feature for molecular spin qubits. Can the authors compute T_1 contribution to T_2 to increase the impact of their study, as T_1 is extensively studied by them in ref [24-28]. As the authors are using full Redfield theory (eq. S1), they can prepare their density from superposition state of their eigen spin Hamiltonian.”

The reviewer raises an important question. The contribution of phonons to T_2 has been overlooked until now and remains an untouched ground. We took on-board the suggestion of the reviewer and performed a new series of spin dynamics simulations to investigate the T -dependence of T_2 .

As suggested by the reviewer, we have now initialized the electronic spin moment in the xy plane. More precisely, we initialized the density matrix to the Boltzmann equilibrium and then applied a $\pi/2$ rotation to the spin magnetic moment.

Let us first discuss the dynamics under the sole effect of the modulation of the g-tensor a source of spin-phonon coupling. As expected, the z-component of the spin, now initialized to 0, recovers its equilibrium value with the same T_1 constant as in the simulations already present in the manuscript. The in-plane spin moment undergoes Larmor precession. By plotting the module of the xy-spin moment we recover a monotonically decay function that can be perfectly fitted with a mono-exponential function. The time constant T_2 obtained by this fitting is found to follow the relation $T_2 \leq T_1$ at any temperature. We note that this result contradicts the commonly accepted result $T_2 = 2 T_1$ obtained by the Redfield theory. However, it should be reminded that the canonical relation $T_2 = 2 T_1$ comes from a second-order time-dependent perturbation theory with linear spin-bath coupling. The simulations presented in this work are instead obtained with a quadratic coupling. This introduces one additional decoherence channel, namely pure spin dephasing, not present in the one-phonon dynamics.

On a more quantitative ground, Redfield relaxation theory predicts

$$\frac{1}{T_2} = \frac{1}{2T_1} + \frac{1}{T_2^*},$$

where T_2^* is the aforementioned pure dephasing contribution. This relation is true for both linear and quadratic coupling, but in the former case the contribution vanishes.

The explicit expression for the dephasing rate reads

$$\frac{1}{T_2^*} \propto -2V_{aa}V_{bb}G^{1-ph}(0, \omega_\alpha) + V_{aa}V_{aa}G^{1-ph}(0, \omega_\alpha) + V_{bb}V_{bb}G^{1-ph}(0, \omega_\alpha)$$

and

$$\frac{1}{T_2^*} \propto -2V_{aa}V_{bb}G^{2-ph}(0, \omega_\alpha, \omega_\beta) + V_{aa}V_{aa}G^{2-ph}(0, \omega_\alpha, \omega_\beta) + V_{bb}V_{bb}G^{2-ph}(0, \omega_\alpha, \omega_\beta),$$

for one- and two-phonon relaxation at the density matrix second-order time-dependent perturbation theory level, respectively.

The function $G^{1-ph}(\omega, \omega_\alpha) = n_\alpha \delta(\omega - \omega_\alpha) + (n_\alpha + 1) \delta(\omega + \omega_\alpha)$ vanishes for $\omega = 0$ as there are no phonons available at zero frequency. The only exception is represented by the three acoustic branches, which however correspond to rigid translations of the entire crystal and do not couple with spin. As a consequence, the one-phonon contribution to pure dephasing is zero. However, the function $G^{2-ph}(\omega, \omega_\alpha, \omega_\beta) = n_\alpha(n_\beta + 1) \delta(\omega - \omega_\alpha + \omega_\beta)$ does not vanish for $\omega = 0$ and therefore provides an additional contribution. The same result was obtained for the spin dynamics under the influence of hyperfine-mediated spin-phonon interaction.

There is an elegant parallel between this dephasing mechanism and the one due to spin-spin interactions, where the flip-flop spin transitions in the spin bath causes decoherence. Both processes are energy conserving and for that to happen two degenerate degrees of freedom need to exchange energy among them.

Let us now compare this finding with literature. Firstly, it is evident that no experimental evidence is available for the canonical relation $T_2=2T_1$ even at high temperature where the contribution of spin-spin interactions is negligible. Indeed, T_2 is invariably found shorter than T_1 . However, the proportionality factor seems to be variable and a more in-depth experimental study should be performed to quantify this relation. Intriguingly, for NV centers, the relation $T_2=0.5 T_1$ has been experimentally observed (but not explained) after dynamical decoupling had been applied to remove spin-spin decoherence effects entirely. This matter merits a dedicated in-depth study and we postpone a full report of these findings to a follow-up paper. However, we include in the revised version of Fig.5 the simulations of T_2 and a discussion about the phonon-induced pure dephasing contribution to decoherence. The final results of the simulations of T_1 are also shown in Fig.5, which, in the previous version of the paper, reported by mistake not-definitive calculations.

“My next concern is regarding spectral function, eq. S3. This includes no anharmonic term. Is there any specific reason for that? The problem is that one could use any value of σ (smearing effect) to fit their experimental T_1 curve.”

We thank the reviewer for raising this point, which we now briefly address in the methods section of the main manuscript in an attempt to clear this up.

In short, when a full integration of the Brillouin zone is performed, as in this case, the resonance condition with spin is always fulfilled. This is true for both one- and two-phonon processes. However, in the case of one-phonon relaxation involving high-energy phonons (Orbach relaxation in high-anisotropy SMMs), the Bose Einstein population of the resonant phonons decreases exponentially with T , and below some value of T lower-energy non-resonant phonons might become relevant thanks to their linewidth anharmonic broadening.

This scenario does not arise for Raman relaxation. Two-phonon processes can always fulfill both resonance and high-population at the same time. This is due to the fact that a two-phonon resonance condition only requires the energy difference among the two phonons to match the small Zeeman spin splitting and no constraints apply to the energy of the single phonons. As we discuss in our manuscript, relaxation in [VO(TPP)] is mediated by optical phonons of incredibly low energy, therefore guaranteeing plenty of thermal population at any temperature investigated here. As a consequence, the Raman relaxation is not affected by the choice of the smearing function and simulations can safely be converged to the harmonic limit, where the Gaussians are vanishingly narrow, and the number of sampled phonons is infinitely large (see Supplementary Note 3). The reviewer can appreciate all this by Figure S12, where the dependency of the results with respect to the Gaussian broadening is shown to become negligibly small as the Brillouin zone integration is improved. A much larger Brillouin zone sampling might improve even more the results, but it becomes computationally very expensive.

Some of the authors have presented this result on the role of the Brillouin zone sampling and its relation to the phonon linewidth in previous works, see for instance refs [35,39]. A full discussion of this and other technical aspects is also reported in the review chapter [*Spin-Phonon Relaxation in Magnetic Molecules: Theory, Predictions and Insights*, A. Lunghi arXiv:2202.03776 (2022)], now ref 43.

“Lastly, the authors show in Fig 5 that by excluding low energy phonons, one could increase the relaxation time of molecular spin-qubit substantially. Can the authors comment on how to achieve this exclusion of these low energy phonons from synthetic chemistry point of view?”

We thank the Reviewer for the comment, which allows us to further clarify this crucial point of our discussion.

The results on T_1 reported in Fig.5 were obtained by excluding all the phonons with an energy < 6 meV, acoustic modes included. We are aware this is a coarse-grained cut-off, but a selective removal of the modes is not practically possible when dealing with dense phonon spectra as the one displayed by molecular crystals. Such a sharp cut-off cannot be completely obtained through synthetic chemistry. Indeed, low-energy optical phonons can be removed but clearly not acoustic modes (which are nevertheless not efficient in inducing spin relaxation when not mixed with optical ones). Hence, we do not expect synthetic strategies like the one proposed in our paper to yield an effect as strong as the one

reported in Fig.5, which is a sort of superior limit of the effect. However, we have verified withing the gas-phase approximation that by removing the four phenyls from [VO(TPP)] as in Vanadyl porphyrin (an existing complex, see for instance K. Yamashita, *et al.*, *Inorg. Chim. Acta.*, 2016, 439, 173–177) the lowest energy vibrational mode at the Γ point is at about 6.5 meV, a four-fold increase in frequency with respect to [VO(TPP)]. Thus, these results clearly point out an interesting direction.

We have further clarified these points throughout the Results and Discussion sections.

REVIEWERS' COMMENTS

Reviewer #1 (Remarks to the Author):

I thank the authors for explaining the importance of the results more clearly now. I am far more excited about the paper, I think it is now reasonable to publish here!

I have two comments:

There are a lot of different energy units used throughout: MHz, meV, etc. A lot of chemists working in this area use wavenumbers as energy units too, especially for vibrational data. So it's all complicated. I think Fig. 4 would be benefited by either adding another axis that presents the energy in units of wavenumbers.

Figure 7 seems incomplete by overlooking the work of Freedman and coworkers on some of the exact same compounds: DOI: 10.1021/jacs.6b08467

Reviewer #2 (Remarks to the Author):

Report on the revised manuscript (Reviewer 2)

“The critical role of ultra-low energy vibrations in the relaxation dynamics of molecular qubits”

By E.Garlatti et al.

1) The authors responded to all critical points of this reviewer. In addition, dephasing contributions $1/T_2^*$ to the transverse relaxation rate $1/T_2$ were included to demonstrate that the commonly accepted canonical relation $1/T_2 = T_1/2$ is not correct.

2) The key results of this work is Figure 5 which now includes as an inset experimental and computed decoherence times. It is strongly recommended to include the plotted data points as in a digital format as a Table in the ESI.

3) Figure S17 of the original submission was now shifted to the main text (Figure 7) providing a comparison with other VO-based systems. Are there data on the decoherence times for the three systems take for comparison with VO(TPP) ?

4) Supplementary Figure 6 with ID28 vs simulations along Γ -Kz transverse should read Figure 7.

5) Figure 6 depicts the lowest energy $1.5 \text{ eV} = 12.1 \text{ cm}^{-1} \text{ eV}$ mode at the Γ point and two movies in the SI which should be more explicitly explained and labelled. These movies show mostly a rotation. What are the first and second derivatives of g_x, g_y, g_z and A_x, A_y, A_z with respect to that mode? Can the authors please list the same quantities for all Γ point vibrations with energies between 0 and 140 cm^{-1} ?

Reviewer #3 (Remarks to the Author):

I thank the authors for their kind reply, they have satisfactorily addressed all my concerns with additional discussion, clarification in the main text and simulations.

Hence, I can now recommend publication of the paper in its present form.

Reviewer #1

"I thank the authors for explaining the importance of the results more clearly now. I am far more excited about the paper, I think it is now reasonable to publish here!"

We are grateful to the Reviewer for the positive comments on the importance of the results and the clarity of the presentation of our revised manuscript. We are glad to have addressed all the previous remarks and we thank for recommending the manuscript for publication in Nature Communications.

"There are a lot of different energy units used throughout: MHz, meV, etc. A lot of chemists working in this area use wavenumbers as energy units too, especially for vibrational data. So it's all complicated. I think Fig. 4 would be benefited by either adding another axis that presents the energy in units of wavenumbers. "

We thank the Reviewer for this suggestion. Now Fig.4 has an additional axis reporting energy in units of wavenumbers.

"Figure 7 seems incomplete by overlooking the work of Freedman and coworkers on some of the exact same compounds: DOI: 10.1021/jacs.6b08467"

We thank the Reviewer for reporting to us the results on relaxation times in VO-based systems in the paper by C.-J. Yu *et al.*. We have now added them to the comparison with [VO(TPP)]. The four compounds show a temperature dependence very similar to VOPc and to the [VO(dmit)₂] compounds, leaving our conclusion on [VO(TPP)] unaltered. Given that the measurements reported in the Ref.(14) are on frozen solution (not in single crystals as for [VO(TPP)] and the other three compounds) we report this comparison in a new figure, Supplementary Figure 22. In the main text, we recall this new Figure and we have added the paper by C.-J. Yu *et al.* as Ref.(14).

Reviewer #2

"The authors responded to all critical points of this reviewer. In addition, dephasing contributions $1/T_2^$ to the transverse relaxation rate $1/T_2$ were included to demonstrate that the commonly accepted canonical relation $1/T_2 = T_1/2$ is not correct."*

We are glad to have addressed all the previous remarks of the Reviewer.

"The key results of this work is Figure 5 which now includes as an inset experimental and computed decoherence times. It is strongly recommended to include the plotted data points as in a digital format as a Table in the ESI."

As suggested by the Reviewer, experimental data and simulations results on relaxation times are now reported in Supplementary Tables 5-6.

"Figure S17 of the original submission was now shifted to the main text (Figure 7) providing a comparison with other VO-based systems. Are there data on the decoherence times for the three systems take for comparison with VO(TPP)?"

Following this on-point suggestion of the Reviewer, we have now added the comparison of [VO(TPP)] decoherence times with that of the same three systems of Fig.7 in Supplementary Fig. 23

“Supplementary Figure 6 with ID28 vs simulations along Γ -Kz transverse should read Figure 7.”

We have corrected the label of Supplementary Figure 7. We thank the Reviewer for noticing this mistake.

“Figure 6 depicts the lowest energy $1.5 \text{ eV} = 12.1 \text{ cm}^{-1} \text{ eV}$ mode at the Γ point and two movies in the SI which should be more explicitly explained and labelled. These movies show mostly a rotation. What are the first and second derivatives of g_x, g_y, g_z and A_x, A_y, A_z with respect to that mode? Can the authors please list the same quantities for all Γ point vibrations with energies between 0 and 140 cm^{-1} ?”

The Reviewer is right and therefore we report here below the legends for Supplementary Movie 1 and 2 that will be added during the final editing/online publication of the manuscript:

Supplementary Movie 1: Molecular distortions associated to the first phonon mode at the Γ -point for a [VO(TPP)] single crystal.

Supplementary Movie 2: Molecular distortions associated to the first vibrational mode of a [VO(TPP)] isolated molecule (gas-phase calculations).

As suggested by Reviewer, we now report in Supplementary Figure 20 the spin-phonon coupling distribution of the A tensor (panel a) and g tensor (panel b) for phonon modes at the Γ point with low frequencies.

Reviewer #3

“I thank the authors for their kind reply, they have satisfactorily addressed all my concerns with additional discussion, clarification in the main text and simulations.

Hence, I can now recommend publication of the paper in its present form.”

We are grateful to the Reviewer for his/her suggestions, which helped us improve the presentation and the content of our work, and for recommending the manuscript for publication in Nature Communications.